# Sesamin Promotes Osteoporotic Fracture Healing by Activating Chondrogenesis and Angiogenesis Pathways

**DOI:** 10.3390/nu14102106

**Published:** 2022-05-18

**Authors:** Zhengmeng Yang, Lu Feng, Ming Wang, Yucong Li, Shanshan Bai, Xuan Lu, Haixing Wang, Xiaoting Zhang, Yaofeng Wang, Sien Lin, Micky D. Tortorella, Gang Li

**Affiliations:** 1Stem Cells and Regenerative Medicine Laboratory, Department of Orthopaedics & Traumatology, Li Ka Shing Institute of Health Sciences, The Chinese University of Hong Kong, Shatin, Hong Kong SAR, China; 1155097544@link.cuhk.edu.hk (Z.Y.); miles.wong@link.cuhk.edu.hk (M.W.); yucongli@cuhk.edu.hk (Y.L.); shanshan@link.cuhk.edu.hk (S.B.); luxuan@link.cuhk.edu.hk (X.L.); 1155102308@link.cuhk.edu.hk (H.W.); zhangxt53@gmail.com (X.Z.); sienlin@cuhk.edu.hk (S.L.); 2Centre for Regenerative Medicine and Health, Hong Kong Institute of Science & Innovation, Chinese Academy of Sciences, Hong Kong SAR, China; lufeng@link.cuhk.edu.hk (L.F.); yaofeng.wang@hkisi-cas.org.hk (Y.W.)

**Keywords:** sesamin, osteoporosis, fracture healing, estrogen, chodrogenesis, angiogenesis

## Abstract

Osteoporotic fracture has been regarded as one of the most common bone disorders in the aging society. The natural herb-derived small molecules were revealed as potential treatment approaches for osteoporotic fracture healing. Sesamin is a member of lignan family, which possesses estrogenic activity and plays a significant role in modulating bone homeostasis. Our previous study reported the promoting effect of sesamin on postmenopausal osteoporosis treatment. However, the role of sesamin in osteoporotic fracture healing has not been well studied yet. In this study, we further investigated the putative treatment effect of sesamin on osteoporotic fracture healing. Our study indicated that sesamin could activate bone morphogenetic protein 2 (BMP2) signaling pathway and further promotes in vitro chondrogenesis and angiogenesis activities. This promoting effect was abolished by the treatment of ERα inhibitor. In the osteoporotic bone fracture model, we demonstrated that sesamin markedly improves the callus formation and increases the cartilaginous area at the early-stage, as well as narrowing the fracture gap, and expands callus volume at the late-stage fracture healing site of the OVX mice femur. Furthermore, the angiogenesis at the osteoporotic fracture site was also significantly improved by sesamin treatment. In conclusion, our research illustrated the therapeutic potential and underlying regulation mechanisms of sesamin on osteoporotic fracture healing. Our studies shed light on developing herb-derived bioactive compounds as novel drugs for the treatment of osteoporotic fracture healing, especially for postmenopausal women with low estrogen level.

## 1. Introduction

Osteoporosis (OP) and its related bone fracture have been regarded as one of the most common bone disorders in the aging society [1]. In the US, there are more than 1.5 million fracture incidents related to OP annually [2]. Considering the long-term rehabilitation period and high recurrence rate, osteoporotic fracture has become a serious public health concern [3]. The traditional clinical treatment methods for osteoporotic fracture include estrogen and bisphosphonates application. However, several side effects caused by these two medications were observed in recent studies, including callus remodeling delay and carcinoma [4,5,6]. Other interventions including calcium and vitamin D supplement also showed poor treatment outcome as well [7]. Recently, the natural herb-derived small molecules were also revealed as potential treatment approaches. For instance, Salvianolic acid B was found to have anti-OP effects by promoting human MBSCs osteogenesis via ERK signaling pathway [8]. Icaritin also promoted new bone formation by activating cAMP signal pathway in osteoblasts [9,10]. In our previous studies, we discovered that sesamin was a potential drug candidate for OP treatment [11]. However, the treatment effect of sesamin on osteoporotic fracture healing has not been well studied yet.

Previous studies reported that estrogen deficiency accelerates OP progression [12]. Estrogen and its receptors (ERα and ERβ) extensively participate in the regulation of bone and joint metabolism [13]. Estrogen receptor signaling is also highly involved in the whole process of bone fracture healing. At the early stage of fracture healing, estrogen deficiency leads to callus formation defectiveness and chondrocyte maturation impairment. Subsequently, the smaller cartilage zone inside the callus delays endochondral ossification [14]. At the last stage, estrogen deficiency results in delayed and hard callus remodeling and fragile cortical bone layer [4]. Moreover, a low estrogen level induces IL-6 upregulation, which is proved to impede the fracture healing process [15]. Other studies also showed that a high ERα/ERβ ratio is a benefit for callus formation, suggesting ERα the major and critical receptor during fracture healing [16].

As estrogen analogs, phytoestrogen family members have been proved the potential of promoting osteoporotic fracture healing. For example, psoralen and equol were proved to promote OVX animal fracture healing in the pre-clinical studies [17,18,19]. Genistein was also found to improve bone fracture healing by activating ERα signaling pathways and osteoblast maturation [17]. Here we hypothesize that sesamin, a phytoestrogen, could promote osteoporotic fracture healing by activating the ERα signaling pathway. In our research, we studied the promoting effect of sesamin on in vitro angiogenesis and the chondrogenesis process, and proved that sesamin exert its function in an ERα-dependent manner. In in vivo study, we constructed the ovariectomy (OVX) induced OP mice model and sesamin was administrated to assess its effect on long bone fracture healing under the condition of estrogen depletion. Our study may bring a glimmer of light on developing novel therapeutic tools for osteoporotic bone fracture healing.

## 2. Material and Methods

### 2.1. Molecular Docking Study

Based on the conformational similarity between estrogen and sesamin, molecular docking analysis was applied to investigate the binding energy of sesamin on human ERα. The three-dimensional structure of ERα protein was obtained from Protein Data Bank (PDB code 2QZO). The software AutoDock Ver. 1.5.6 (Scripps, La Jolla, CA, USA) was used to convert PDB formatted chemical and protein into PDBQT version for binding pocket characterization. The software Pymol Ver. 2.3.2 (DeLano Scientific LLC, Palo Alto, CA, USA) was used to remove irrelevant hydroxyl and phosphate radicals. The AutoDockTools analysis was conducted for molecular docking. The hydrogen bone interaction was visualized by Pymol. Other residual interactions were analyzed by Discovery Studio software (Dassault Systemes BIOVIA, San Diego, CA, USA).

### 2.2. Cell Transient Transfection and Dual-Luciferase Reporter Assay

The pGL4 luciferase vector harboring human BMP2 promotor sequence was constructed by GenePharma (Shanghai, China). The Renilla luciferase vector was obtained from Cell Signaling Technology (Albany, NY, USA). Lipofectamine 3000 (Thermo Fisher Scientific, Waltham, MA, USA) reagent was used for transfection according to the manufacturers’ instruction. The human bone marrow mesenchymal stem cells (BMSCs) were isolated and kept in our laboratory as previously described [20]. In general, the human BMSCs were seeded into a 12-well plate and grown until 70–80% confluence. For transfection, 1 μg plasmid DNA with 1 μL Lipofectamine^TM^ 3000 were mixed in 500 μL Opti-MEM^TM^ (Thermo scientific, Waltham, MA, USA), and added into the culture medium. After 12-h incubation, cells were treated with 0.5 μM sesamin (59867, Sigma-Aldrich, St. Louis, MO, USA), 100 nM estradiol (E8875, Sigma-Aldrich, St. Louis, MO, USA), 0.2 nM amcenestrant (Sigma, St. Louis, MO, USA) or the combinations for an additional 24 h. For dual-luciferase assay, Dual-Luciferase^®^ Reporter Assay Kit (Promega, Madison, WI, USA) was applied as mentioned before [20]. The luciferase activity was measured by PerkinElmer VictorTM X2 2030 multilateral reader (PerkinElmer, Waltham, MA, USA). The firefly luciferase activity was normalized to renilla luciferase activity.

### 2.3. Cell Viability Assay

Human umbilical vein endothelia cells (HUVECs) were purchased from ATCC and cultured with Dulbecco’s Modified Eagle Medium (DMEM) supplemented with 10% FBS and 1% PSN. The cytotoxicity of sesamin on human BMSCs and HUVECs was evaluated by 3-(4,5-Dimethylthiazol-2-yl)-2,5-diphenyltetrazolium bromide (MTT) assay as described before [21]. Briefly, BMSCs or HUVECs were seeded in a 96-well plate at the density of 5 × 10^3^/well and cultured for 24 h. The cells were then supplemented with sesamin at various concentrations for another 72 h. After that, 10 μL of 0.5 mg/mL MTT was added into each well and incubate at 37 °C for an additional 4 h. After incubation, the culture medium was removed and 100 μL DMSO was added for the color visualization. The light absorbance was assessed by a 96-well plate reader (Molecular Devices, San Jose, CA, USA) at the wavelength of 570 nm.

### 2.4. Capillary Tube Formation Assay

Matrigel Matrix (Corning, New York, NY, USA) was pre-cooled by ice. One hundred microliters of Matrigel Matrix was loaded into the wells of a 96-well plate to prepare the basement membrane. HUVECs were seeded above the Matrigel Matrix layer in a 96-well plate at the density of 4 × 10^4^ cells/well. After 1-h incubation, sesamin was applied to the cells to the final concentration of 0.5 μM. After 3- and 6-h treatment, tubular structures were photographed by inverted phase-contrast microscopy (Nikon, Tokyo, Japan). The tube formation degree was evaluated by measuring total branching length and number of junctions using ImageJ software (NIH, Bethesda, MD, USA).

### 2.5. Chondrogenic Induction

A three-dimensional cell culture system was applied for chondrogenesis assay as described before [22]. Human BMSCs were resuspended at the density of 10^6^ cells/10 μL in 10 μL StemPro^TM^ Chondrogenesis Differentiation medium (A1007101, Thermo Fisher, Waltham, MA, USA). The 10-μL droplet was added in the center of the culture wells. After 30 min, another 1 mL chondrogenic induction medium was added to the culture well. The cell spherical aggregate formed subsequently. The estradiol (100 nM) or sesamin (0.5 μM) was supplemented into the medium 3 days after seeding. After 21-day induction, all cell pellets were collected and fixed with 10% formalin for 24 h. Fixed pellets were then dehydrated with 30% saccharose dissolved in formalin. Optimal cutting temperature compounds (OCT, Sakura Finetek, St. Torrance, CA, USA) were used for sample embedding and 7-μm sections were cut by cryostat (Leica, Wetzlar, Germany). Alcian blue staining was applied according to our previous work [21].

### 2.6. Real-Time PCR

Total RNA was extracted from cultured cells by TRIzol reagent (Life Technologies, Carlsbad, CA, USA). Complementary DNA was reversely transcribed by using reverse transcriptase (Takara, Otsu, Japan). Real-time PCR was applied by applying SYBR Green Mastermix (Thermo Fisher, Waltham, MA, USA) and the result was visualized by ABI 7500 Sequencing Detection System (Applied Biosystems, Waltham, MA, USA). The expression level of target genes was calculated by 2^−^^ΔΔCq^ method and GAPDH was used as the internal control. The primers were designed according to the CDS sequence of genes as listed: GAPDH (forward: 5′-CCT CGT CTC ATA GAC AAG ATG GT-3′, reverse: 5′-GGG TAG AGT CAT ACT GGA ACA TG-3′), VEGF (forward: 5′-GAA GTG GTG AAG TTC ATG GAT GTC-3′; reverse: 5′-CGA TCG TTC TGT ATC AGT CTT TCC-3′), Mmp2 (forward: 5′-CCT GTT TGT GCT GAA GGA-3′; reverse: 5′-CAA GAA GGG GAA CTT GCA-3′), Mmp9 (forward: 5′- TTC ATC TTC CAA GGC CAA TC-3′; reverse: 5′-CTT GTC GCT GTC AAA GTT CG-3′), and Mmp14 (forward: 5′-GCA GAA GTT TTA CGG CTT GCA-3′; reverse: 5′-TCG AAC ATT GGC CTT GAT CTC-3′).

### 2.7. Western Blot Analysis

Total protein was obtained by lysing the cells with RIPA buffer (Sigma-Aldrich, St. Louis, MO, USA) supplemented with protease inhibitor cocktail (Roche, Basel, Switzerland). SDS-PAGE gel was used for electrophoresis, and total protein was then electroblotted onto the polyvinylidene fluoride (PVDF) membrane. Five percentage non-fat milk was applied to the PVDF membrane for blocking. After that, primary antibodies were applied to the membrane including mouse anti-β-catenin (1:3000, 610153, BD Biosciences, San Jose, CA, USA), rabbit anti-VEGF (1:2000, ab32152, Abcam, Cambridge, UK), rabbit anti-Mmp2 (1:2000, ab92536, Abcam, Cambridge, UK), rabbit anti-Mmp9 (1:2000, ab76003, Abcam, Cambridge, UK), rabbit anti-Bmp2 (1:3000, ab284387, Abcam, Cambridge, UK), rabbit anti-Sox9 (1:1000, ab185230, Abcam, Cambridge, UK), and rabbit anti-Aggrecan (1:2000, ab36861, Abcam, Cambridge, UK). The results were visualized by using Kodak film developer (Fujifilm, Tokyo, Japan) and quantified by ImageJ software (NIH, Bethesda, MD, USA).

### 2.8. Osteoporotic Bone Fracture Animal Model

Animal experiments were authorized under the Ethics Committee approval of the Chinese University of Hong Kong (Ref: 19-276-HMF). In general, 36 female C57BL/6 mice (20–25 g) were randomly subjected to sham (*n* = 12) or ovariectomy (*n* = 24) groups. The mice OP was induced by ovariectomy (OVX) surgery. Briefly, the OVX was conducted by spaying the bilateral ovaries as mentioned previously [11]. The sham surgery was performed by repeating the same procedure except for ovary removing. After 1-month OP induction, all the mice were anesthetized. A 25-gauge needle was inserted into the right femur as an internal fixator. The fracture was created by a guillotine-like fracture device as described before [23]. After 3-day recovery, the osteoporotic fracture mice were randomly divided into two groups, each group were administrated with saline (100 μL/day) or sesamin (100 μL, 80 mg/kg/day) by oral gavage for another 14 or 28 days. At the endpoint, all animals were sacrificed by overdose anesthesia. Both fractured and non-fractured femurs were collected for further analysis.

### 2.9. Mechanical Testing and Micro-Computed Tomography Analysis

To evaluate the fracture healing condition, radiographic photos of mice femur fracture sites were taken by Faxitron Bioptics X-ray system (Faxitron, Tucson, AZ, USA) at both day 14 and 28 post-fracture. The high-resolution μCT (Scanco Medical, Wangen, Switzerland) was used for three-dimensional reconstruction according to our previous protocol (threshold σ = 1.2, support = 2) [24]. The low and high mineral density of the newly formed callus were identified by different thresholds (110 for low, 158 for high). Mechanical testing was performed by using a 3-point bending machine (Tinius Olsen, Redhill, UK) according to our previous protocol [24]. The fractured right femur was put at anterior-posterior position with two supporters. The loading point was defined as the center of the fracture site. The parameters including elasticity modulus, ultimate load, and energy to failure were calculated based on our previous work.

### 2.10. Histological Studies

The right femurs were fixed by 10% formalin for 24 h and decalcified by 10% EDTA at 37 °C for about 14 days. The dehydration and paraffin embedding processes were then performed as described previously. After that, the paraffin block was sectioned into 7-μm thickness. Safranin O/Fast green and alcian blue/nuclear fast red staining was performed according to the previous study [25]. The cartilage area and new formed sponge bone tissue area were measured by Image J (NIH, Bethesda, MD, USA)

### 2.11. Immunohistochemistry and Immunofluorescence Staining

Immunohistochemistry (IHC) and immunofluorescence (IF) staining of cell pellet and femur tissue sections were performed as described before [26,27]. The sections were incubated with primary antibodies including rabbit anti-BMP2 (1:1000, ab284387, Abcam, Cambridge, UK), rabbit anti-SOX9 (1:500, ab185230, Abcam, Cambridge, UK), and rabbit anti-CD31 (1:1000, ab182981, Abcam, Cambridge, UK). The secondary antibodies used for IF staining included goat anti-rabbit IgG H&L (Alexa Fluor Fluor^®^ 488 and 647) (1:1000, ab150077&ab150083, Abcam, Cambridge, UK). The secondary antibody for IHC staining was goat anti-rabbit IgG H&L (HRP) (1:1000, ab6721, Abcam, Cambridge, UK). The signal visualization of the IHC staining was developed by horseradish peroxidase-streptavidin system (Dako, Santa Clara, CA, USA). Photographs of the selected areas were taken under a light microscope (Leica, Wetzlar, Germany).

### 2.12. Statistical Analysis

All data are calculated as mean ± standard error of the mean. Student *t*-test (Unpaired, two-tailed) was applied for parametric data and Mann–Whitney U test was applied to analyze non-parametric data. The calculation was performed under GraphPad Prism 8 (San Diego, CA, USA). The difference was considered as significant when *p* < 0.05.

## 3. Result

Sesamin was predicted to be ERα ligand and promoted BMP2 promoter activity in an ERα-dependent manner.

Sesamin is a member of lignan family and is considered as a type of phytoestrogen, which is suggested to possess estrogenic activities (Figure 1A). Prediction of possible interaction between sesamin and the ligand-binding pocket (LBD) of ERα was performed using the molecular docking method. The predicted interaction structural model was obtained and visualized by Pymol software. The result indicated that sesamin could form a conventional hydrogen bond with the residue ARG346 of ERα (Figure 1B). However, only hydrogen bond interactions could be observed by Pymol, which could not explain the calculated binding free energy. Therefore, the Discovery Studio software was applied to investigate the existence of other putative interactions between sesamin and residues of ERα LBD (Figure 1C). The dual-luciferase reporter assay results indicated that both sesamin and estradiol could upregulate BMP2 promotor activity in BMSCs. While these positive effects were attenuated by the co-treatment of ERα specific inhibitor amcenestrant (Figure 1D). The result indicated that sesamin promoted BMP2 promoter activity in an ERα-dependent manner.

### 3.1. Sesamin Promoted Tube Formation of HUVECs through ERα

The cytotoxicity test result indicated that sesamin had no obviously toxic effect on HUVECs at the concentration ranging from 0 to 10 μM (Appendix A). The endothelial cell tube formation assay was performed to investigate the effect of sesamin on angiogenesis. The microscopic image revealed that both sesamin and estradiol could significantly promote in vitro tube formation activity, while the promoting effect of both estradiol and sesamin were ameliorated by the application of the ERα inhibitor (Figure 2A). Statistical analysis of total branching length (Figure 2B) and number of junctions (Figure 2C) further confirmed that both estradiol and sesamin exert their pro-angiogenesis effects by activating ERα. Western blot analysis results indicated that both estradiol and sesamin upregulated the protein expression level of angiogenesis markers including VEGF, Mmp2 and Mmp9 in HUVECs. While this promoting effect was also ameliorated by the application of ERα inhibitors (Figure 2D,E). Real-time PCR analysis of angiogenesis marker mRNA expression level further confirmed our conclusions (Appendix A).

### 3.2. Sesamin Indued Chondrogenesis by Activating ERα and BMP2 Signaling

We further tested the effect of sesamin on promoting BMSCs chondrogenesis. The MTT assay result indicated that sesamin showed no apparent toxicity on BMSCs at the concentration from 0 to 10 μM (Appendix A). Alcian blue staining of chondrogenic pellet revealed that both estradiol and sesamin could promote proteoglycans synthesis. However, the promoting effects of estradiol and sesamin were both abolished by the administration of ERα inhibitor (Figure 3A). The IHC and IF staining results indicated that the protein expression level of chondrogenesis markers including Col II (Figure 3B), SOX9 (Figure 3C) and BMP2 (Figure 3D) were increased by both estradiol and sesamin treatment while reversed by amcenestrant co-treatment. Western blot analysis confirmed the expression pattern of SOX9 and BMP2 level were in compliance with the IF result (Figure 3E,F), which further strengthened our conclusion that sesamin regulated BMSCs chondrogenesis by activating ERα signaling and the downstream BMP2 and SOX9 expression.

### 3.3. Sesamin Oral Administration of Improved Osteoporosis Fracture Healing In Vivo

The fracture site healing conditions were evaluated by X-ray and micro-CT analysis 14 and 28 days after surgery. The X-ray images indicated the OVX group had relatively small calluses compared with NC group at day 14. While sesamin administration slightly reversed this tendency. On day 28, the fracture gaps were all filled in each group. However, a weaker bridge conjunction at the fracture site was observed in the OVX group, while sesamin treatment could improve the conjunction (Figure 4A). Micro-CT reconstruction images further illustrated that at day 14 post-fracture, the fracture line of sham group was almost merged. However, an obvious gap still exists at the fracture site in the OVX group. The sesamin administration group showed a relative narrower gap and larger callus compared with the OVX group. At day 28, there was still no connections formed in the OVX group, while the sesamin treatment improves the fracture reconnection (Figure 4B). The parameters including bone volume/total volume (BV/TV) at the fracture sites at day 14 (Figure 4C) and 28 (Figure 4D), as well as callus area at day 28 (Figure 4E) post-surgery also demonstrated the promoting effect of sesamin on fracture healing compared with OVX group. Furthermore, the results of biomechanical analysis at day 28 post-fracture showed that the E-modulus, ultimate load as well as energy to failure, is obviously ameliorated within the sesamin treatment group (Figure 4F).

### 3.4. Sesamin Facilitated Osteoporotic Fracture Healing by Promoting Chondrogenesis at Early Stage

To better evaluate the promoting effect of sesamin on osteoporotic fracture healing, a histological analysis was performed to further study the fracture healing condition. At day 14 post fracture, Safranin O/fast green and alcian blue/Nuclear fast red staining revealed that the OVX group showed a relatively less proteoglycan formed in new-formed callus area compared with the sham group, whereas the sesamin treatment could significantly promote cartilage formation (Figure 5A1–A6). At day 28 post-fracture, more sponge tissue was observed in the OVX group compared with the sham group. Less sponge tissue and better bone reconnection condition was observed in the sesamin treatment group (Figure 5B1–B6). Moreover, IHC staining results showed that BMP2 and SOX9 expression were repressed in the OVX group, while the sesamin administration manifestly rescued their expression 14 (Figure 5A7–A12) and 28 days (Figure 5B7–B12) after surgery, respectively. The semiquantitative analysis result also indicated that sesamin treatment could alleviate OVX induced cartilage tissue decrease and promote new bone formation upon fracture points (Figure 5C,D).

### 3.5. Sesamin Ameliorates Angiogenesis within Fracture Healing of OVX Mice

The stimulating effect of sesamin on angiogenesis during osteoporotic fracture healing was also studied in the OVX mice model. The IF staining result showed that the positive staining of angiogenesis marker CD31 in the bone fracture area was decreased significantly in the OVX group compared with the NC group at day 14 and 28 post-surgery. However, the sesamin treatment greatly recovered the CD31 expression (Figure 6A,B). The semiquantitative analysis also confirmed our conclusion (Figure 6C).

## 4. Discussion

Over the past decades, enormous attempts have been devoted to ameliorate hormone-related bone degeneration [28]. As one of the potential therapeutic approaches, small molecules have their special advantages, including higher stability and structural integrity, lower immunogenicity, and more commercial affordability compared with protein-based therapeutic strategies [29]. In the current study, we proved that a small molecule, sesamin could promote in vitro angiogenesis and chondrogenesis in an ERα-dependent manner. In an in vivo fracture healing study, more vigorous callus formation and a larger cartilaginous area were observed at the early-stage fracture healing site in the sesamin treatment group compared to the OVX group; while sesamin administration also led to a narrower fracture gap, larger callus and higher angiogenesis at the late-stage fracture healing sites. These findings further supported our conclusions that sesamin could significantly facilitate osteoporotic fracture healing by activating chondrogenesis and angiogenesis under estrogen deficiency condition.

Sesamin was identified as a member of the lignan families and shared a high structural similarity to estradiol, an estrogen steroid hormone and the major female sex hormone. As reported before, estrogen played a significant role in bone metabolism. Previous studies showed that estrogen deficiency resulted in increased bone resorption and bone microstructure damage in postmenopausal women [30]. Clinically, estrogen has been commonly used for postmenopausal women to attenuate bone loss by decreasing or slowing bone turnover rates. However, supplementation of estrogen or its replacement showed severe side effects in clinical application, including potential risk of vaginal bleeding, breast cancer, uterine cancer, and cardiovascular events [31]. Therefore, dietary and herbal approaches have been introduced as alternative approaches. As a phytoestrogen, sesamin showed strong repressive effects against breast cancer by inhibiting proliferation and inducing apoptosis of breast cancer cells [32], which demonstrated a high clinical safety compared with estrogen and other replacement. On the other hand, the effect of estrogen on the proteoglycan metabolism of chondrocytes was still under controversy. Scranton et al. observed that high estradiol application suppressed DNA synthesis in human chondrocytes and restrains proteoglycan synthesis [33]; while Moslemi’s group found that estrogen could promote glycosaminoglycan synthesis in rabbit joint chondrocytes [34]. The contradictory results may be due to the cell line, cell stage, and estrogen dosage differences applied in different studies, and further hindered the clinical availability of sesamin on promoting chondrogenesis and followed endochondral ossification during fracture healing. In our study, we proved that sesamin promoted chondrogenesis both in vitro and in vivo in the process of fracture healing. Sesamin was also found to enhance GAGs synthesis and alleviate IL-1β induced inflammation within human chondrocytes [35], which further supported our conclusion that sesamin supplement facilitated the endochondral ossification process during osteoporotic fracture healing.

In our study, we found that sesamin regulated chondrogenesis and angiogenesis by activating ERα receptor. In recent years, many studies have demonstrated the role of ERα in bone regeneration. It was reported that the bone mineral density (BMD) was severely reduced in ERα knockout mice [36]. The ER single- or double-knockout studies further indicated that ERα was mainly responsible compared with ERβ for the OVX induced high RANKL/OPG ratio and low bone turnover rate [37]. Furthermore, Lee et al. revealed that bone adaptation to mechanical loading highly relied on ERα expression. ERα deficiency was considered as one of the reasons for postmenopausal women’s high fracture rate [38]. In the fracture healing process, ERα deficiency may severely impair the interaction between osteoclasts and osteoblasts, and subsequently affected callus remodeling [39]. These findings indicated the significance of ERα in bone metabolism. In our research, molecule docking results showed the free energy of the association between sesamin and ERα with less than −5.0 kcal/mol, suggesting a desirable binding affinity between them. The luciferase assay result indicated that amcenestrant, a specific ERα inhibitor, could significantly repress sesamin induced BMP2 promoter activity. Amcenstrant application also ameliorated the promoting effect of sesamin on chondrogenesis and angiogenesis in vitro. Taken together, we summarized that sesamin could activate ERα signaling, and further modulate the downstream bone remodeling process.

Sesamin was found to regulate the expression of chondrogenesis marker BMP2 and SOX9, as well as angiogenesis markers including VEGF and MMPs in an ERα dependent manner. Gazit’s group characterized the ERα response element in the BMP2 promoter region, which could be activated by estradiol binding and increase the BMP2 transcription [40]. In our study, we also demonstrated that both sesamin and estradiol could activate the BMP2 promoter activity via ERα, suggesting a ERα-related regulating mechanism of sesamin on BMP2. Previous studies have determined a significant role of BMP signaling in bone development and fracture healing [41,42]. Mesenchymal progenitor cells at the mice bone repair site did not differentiate unless BMP2 existed, suggesting the significance of BMP2 at the initiation stage of bone fracture healing [43]. Research on chondrocyte- and osteoblast-specific BMP2 conditional knockout mice indicated that BMP2 deficiency leads to a delayed cartilage callus formation and damaged mineralization process, which revealed that BMP2 deeply participates in the endochondral ossification process of fracture healing [44]. Furthermore, Jin’s study found that BMP2 promoted chondrogenesis via activating the p38/MAPK signal pathway and inhibiting SOX9 degradation [45]. BMP2 could also promote in vitro angiogenesis by association with VEGF and FGF2 [46]. All these previous studies supported our finding that sesamin significantly activated BMP2 expression via ERα signaling, and further regulated the downstream SOX9 and VEGF expression as well as chondrogenesis and angiogenesis in vitro and in vivo, respectively.

In our previous study [21], we used a sesamin dosage of 80 mg/kg/day and 160 mg/kg/day to investigate the effects on osteoporosis development, and both 80 and 160 mg/kg/day sesamin had anti-osteoporosis effects. We did not observe any toxicity effects of sesamin in our studies, as well as in the published literature. In the current study, we used 80 mg/kg/day concentration and showed that sesamin promoted osteoporotic fracture healing. We understand that sesamin 80 mg/kg/day dosage was still quite high for clinical use, and future studies on developing small molecular drugs mimicking sesamin structure may be needed to reduce the costs and make it safer.

In conclusion, our work showed sesamin promoted osteoporosis bone fracture healing by activating angiogenesis and chondrogenesis during endochondral ossification and fracture callus formation. Sesamin exerted its regulating functions via activating ERα and BMP2 signaling pathways and their downstream genes. This study elucidates a new mechanism of sesamin on promoting effects of osteoporotic fracture healing, especially for postmenopausal women with low estrogen levels. However, despite sesamin being considered a safe natural molecule, its long-term safety profile and dosage still need careful investigation and small molecule drugs mimicking the sesamin structure may be the future research endeavors.

## Figures and Tables

**Figure 1 nutrients-14-02106-f001:**
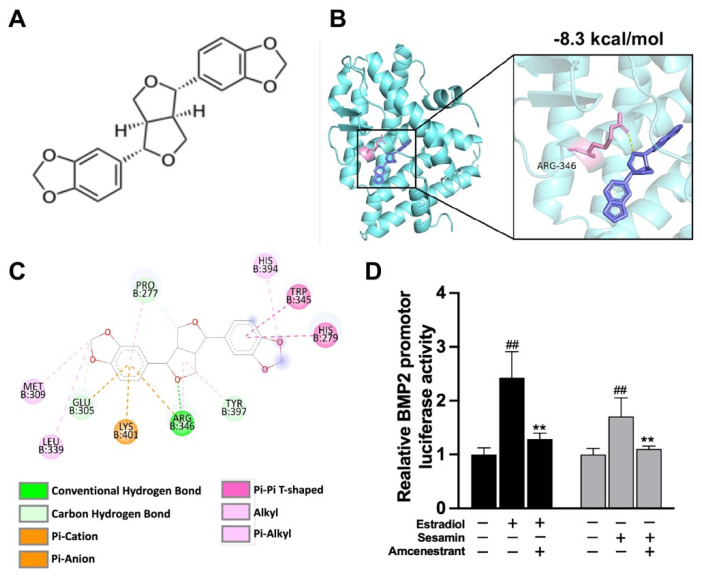
Sesamin exhibited high binding affinity to Erα: (**A**) Molecular structure of sesamin. (**B**) Three-dimensional structure of the predicted interaction structural model between sesamin and the ligand binding pocket of ERα. Sesamin was violet colored. Side chains in ERα were pink colored (**C**) Predictions of various binding forces between sesamin and residuals of ligand binding pocket of ERα. (**D**) Luciferase activity of BMP2 promotor reporter system upon treatment of sesamin, estradiol and ERα inhibitor amcenestrant in vitro (*n* = 6; ^##^
*p* < 0.01 vs. NC group; ** *p* < 0.01 vs. non-amcenestrant treatment group).

**Figure 2 nutrients-14-02106-f002:**
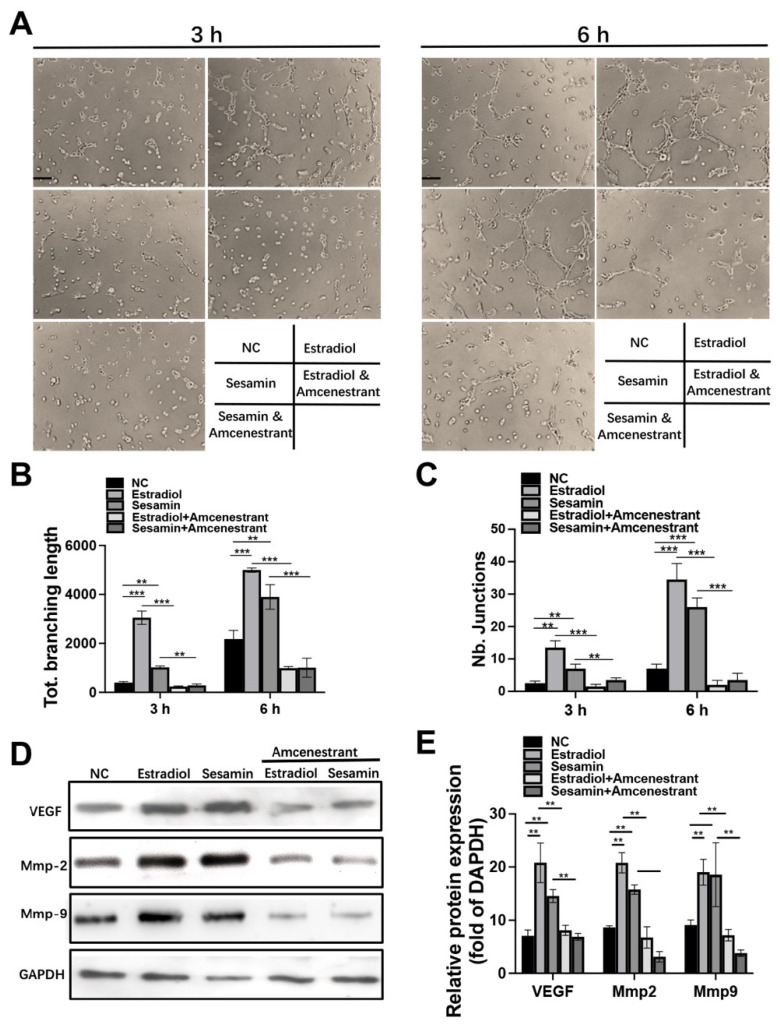
Sesamin promoted HUVECs angiogenesis in vitro: (**A**) Images of tube formation at 3 h and 6 h post-induction with the treatment of different combinations. (**B**,**C**) Semiquantitative analysis of total branching length of each group (**B**) and number of junctions (**C**) in each group. (**D**,**E**) Western blot (**D**) and semiquantitative analysis (**E**) of angiogenesis markers expression 6 h after angiogenesis induction (*n* = 6; ** *p* < 0.01, *** *p* < 0.001).

**Figure 3 nutrients-14-02106-f003:**
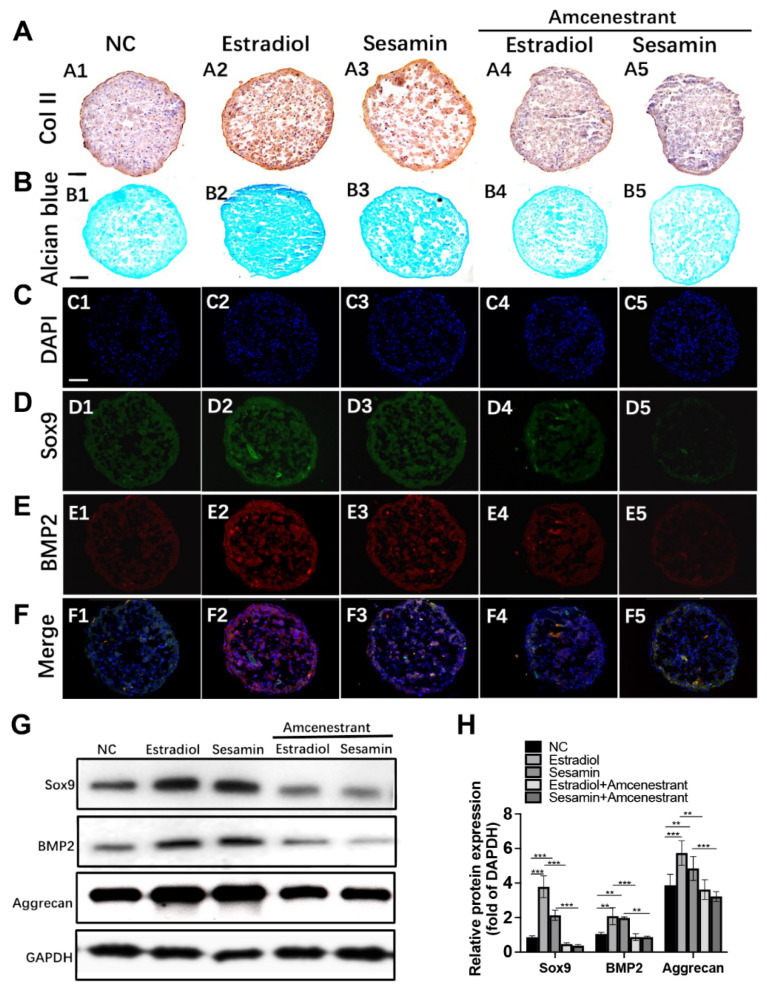
Sesamin promoted chondrogenesis of BMSCs in vitro: (**A**) IHC staining of BMSCs cell pellets under different treatments using Collagen type II antibody. (**B**) Alcian blue staining of cell pellets. (**C**–**F)**. The IF staining of cell pellets using SOX9 (**D**) and BMP2 (**E**) antibodies. DAPI was used as a negative control (**C**). The figures were merged (**F**). (**G**,**H**) Western blot (**G**) and semi-quantitative analysis (**H**) of chondrogenesis markers expression after 21-day chondrogenesis induction (*n* = 6; ** *p* < 0.01, *** *p* < 0.001).

**Figure 4 nutrients-14-02106-f004:**
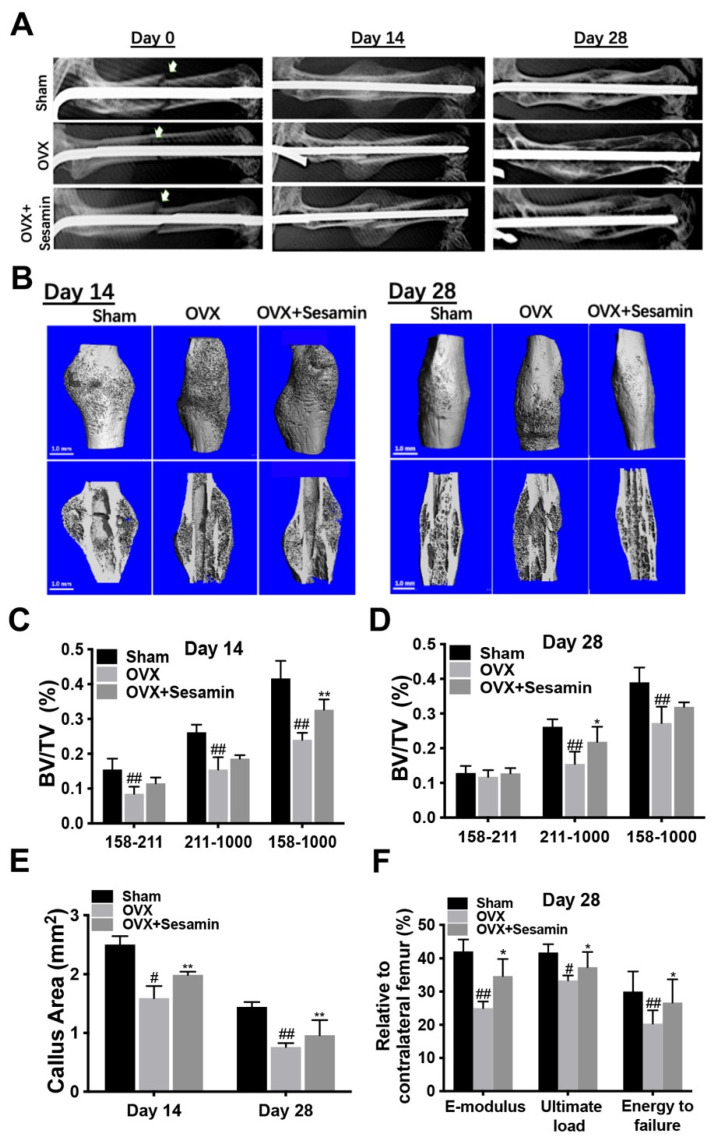
Sesamin promoted osteoporosis fracture healing in vivo: (**A**) X-ray images of the fracture femurs at different time points (the fracture sites were indicated by white arrows). (**B**) Three-dimensional reconstruction images of new formed callus 14 or 28 days after surgery. (**C**,**D**) Bone volume/tissue volume (BV/TV) in each group at day 14 (**C**) and 28 (**D**) post-surgery. (**E**) Measurements of callus area at different timepoints after surgery. (**F**) Analysis of biomechanical parameters including E-modulus, ultimate load and energy to failure of fractured femurs normalized to non-fractured femurs at day 28 post-surgery (*n* = 6; ^#^
*p* < 0.05, ^##^
*p* < 0.01 vs. Sham group; * *p* < 0.05, ** *p* < 0.01 vs. OVX group).

**Figure 5 nutrients-14-02106-f005:**
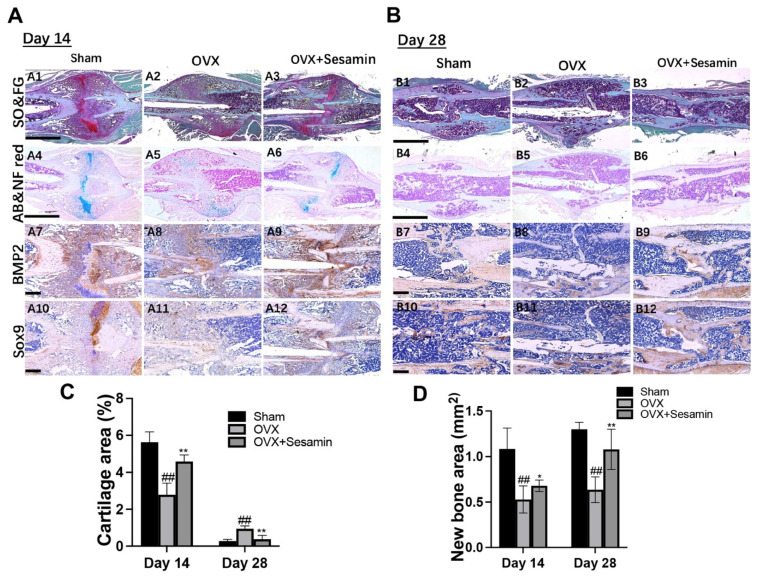
Histological analysis of osteoporosis fracture healing upon sesamin treatment in vivo. (**A**,**B**) Safranin O/fast green (SO and FG) and alcian blue/nuclear fast red (AB/NF red) staining of femur fracture sites 14 (A1–A6) and 28 days (B1–B6) after surgery. IHC staining of femur fracture sites using BMP2 and SOX9 antibodies at day 14 (A7–A12) and day 28 (B7–B12) post-surgery. (**C**) Semi-quantitative analysis of cartilage area of each group at different timepoints after fracture. (**D**) Semi-quantitative analysis of newly formed sponge bone area in each group (*n* = 6; ^##^
*p* < 0.01 vs. Sham group; * *p* < 0.05, ** *p* < 0.01 vs. OVX group).

**Figure 6 nutrients-14-02106-f006:**
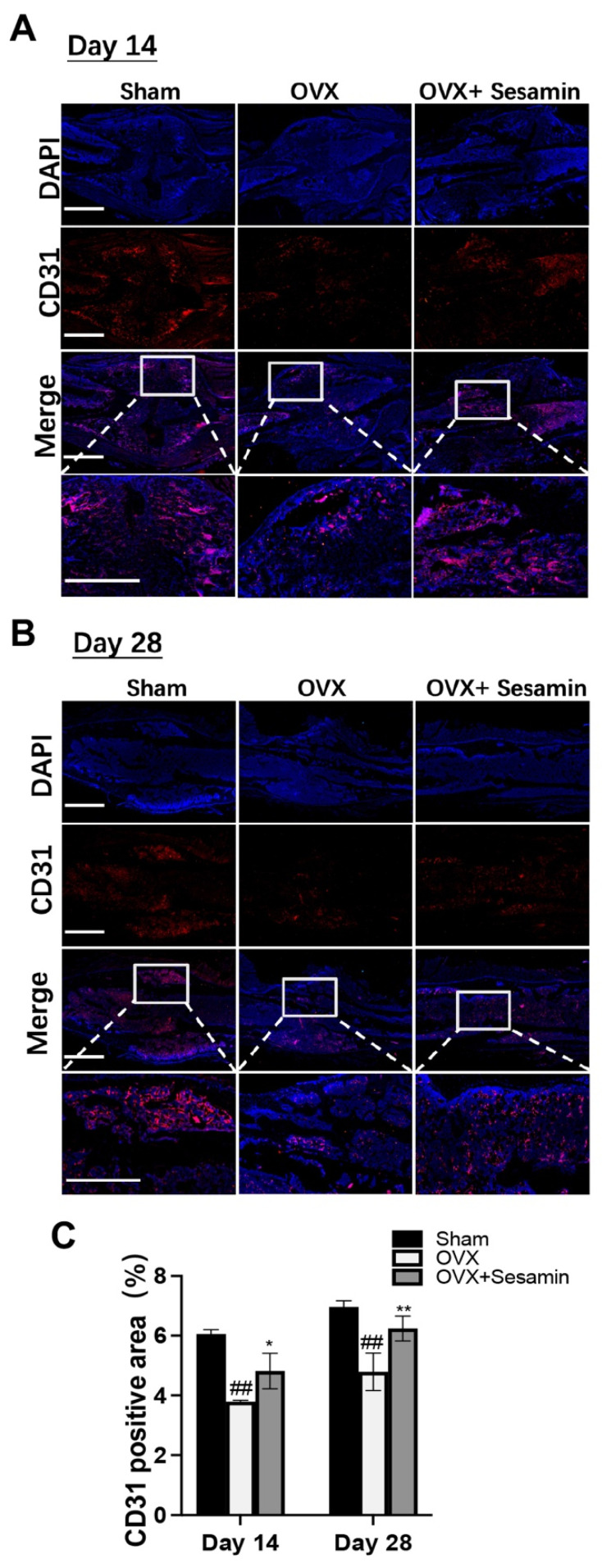
Sesamin promoted angiogenesis within callus area in vivo: (**A**,**B**) The IF staining of CD31 positive area in each group 14 (**A**) or 28 days (**B**) after fracture. (**C**) Semi-quantitative analysis of CD31 positive area (*n* = 6; ^##^
*p* < 0.01 vs. Sham group; * *p* < 0.05, ** *p* < 0.01 vs. OVX group).

## Data Availability

Not applicable.

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
