# Peer review of "Sesamin Promotes Osteoporotic Fracture Healing by Activating Chondrogenesis and Angiogenesis Pathways"

_nutrients, 2022, doi:10.3390/nu14102106_

Round 1

Reviewer 1 Report

The authors hypothesize that sesamin, a phytoestrogen, could promote osteoporotic fracture healing by activating ERα signaling pathway. In our research, we studied the promoting effect of sesamin on in vitro angiogenesis and chondrogenesis process, and proved that sesamin exert its function in an ERα-dependent manner. In in vivo study, we constructed the ovariectomy (OVX) induced OP mice model and sesamin was administrated to assess its effect on long bone fracture healing under the condition of estrogen depletion. In conclusion, sesamin promoted osteoporosis bone fracture healing by activating angiogenesis and chondrogenesis during endochondral ossification and fracture callus formation. Sesamin exerted its regulating functions via activating ERα and BMP2 signaling pathways and their downstream genes. This study elucidates a new  mechanism of sesamin on promoting effects of osteoporotic fracture healing, especially for postmenopausal women with low estrogen level.

The introduction is clear, stating the role of estrogens in the formation of the fracture callus. In the final part, it indicates that phytoestrogens such as sesamin can intervene

The methodology is exhaustively described and other laboratories can repeat the studies.

The results are described in a clear and orderly way following a physiological scheme that allows us to understand the process.

The discussion is clear adjusting to the results obtained.

Author Response

Thank you very much for your appreciation.

Reviewer 2 Report

This paper is innovative and It could open new windows on the current research on the putative treatment effect of sesamin on osteoporotic fracture healing. I have very minor comments since the paper is well designed and the results are really cohomprensive.

1) this study established 80 mg/kg/day as treatment dosage. There are similar studies at different dosages? could you better explain the reason of this dosage?

2) what about the side effects about the sesamin reported in this study? a safety dependent dosage has not been discussed and this must be raised

3) how many mices completed this trial? there was any adverse outcome? please describe better this matter

Author Response

(1) this study established 80 mg/kg/day as treatment dosage. There are similar studies at different dosages? could you better explain the reason of this dosage?

Response: In our previous study (doi: 10.3390/nu13124455), we have used sesamin dosage of 80 mg/kg/day and 160 mg/kg/day to investigate the effects on osteoporosis development, and both 80 and 160 mg/kg/day sesamin had anti-osteoporosis effects. We did not observe any toxicity effects of sesamin in our studies, as well as in the published literatures.  In the current study we used 80 mg/kg/day concentration and showed that sesamin promoted osteoporotic fracture healing.  We understand that sesamin 80 mg/kg/day dosage was still quite high for clinical use and future studies on developing small molecular drugs mimicking sesamin structure may be needed to reduce the costs and make it safer. We have added these statements in the revised manuscript (line 422-429).

(2) what about the side effects about the sesamin reported in this study? a safety dependent dosage has not been discussed and this must be raised

Response: Thank you for your comments. We did not observe any toxicity effects of sesamin in our studies (dosage at 80 and 160 mg/kg/day in mice), as well as in the published literatures. In our studies, no abnormality was found in body weight and organs in the mice. The concentration of sesamin (80 mg/kg/day) in the current study was determined based on our previously published papers and cytotoxicity tests (MTT), and we did not observe any toxic effects. We have added discussion of the revised manuscript (line 435-438).

(3) how many mice completed this trial? there was any adverse outcome? please describe better this matter

Response: Thank you for your comments. We have used total 36 mice in this study. As we discussed above, sesamin is very safe and we haven’t found any side effects in the current study.  The body weight measurement and gross observation of all major organs of the mice did not show any difference following the sesamin systemic treatment in our study.  We have added these statements in the revised manuscript (line 422-429).